# The Evaluation and Promotion Path of Green Innovation Performance in Chinese Pollution-Intensive Industry

**Caiming Wang [1] and Jian Li [1,2,*]**

[1]   School of Management, Tianjin University of Technology, Tianjin 300384, China; 153103403@stud.tjut.edu.cn

[2]   School of Management and Economics, Tianjin University, Tianjin 300072, China

*   Correspondence: wangzhixiu@tju.edu.cn

**Abstract:** Innovation driven green development has become the key to realizing the transformation and upgrading of pollution-intensive industries and the improvement of economic quality and efficiency in the new era. Based on the identification of pollution-intensive industries, this study evaluated the green innovation performance of Chinese pollution-intensive industry from 2014 to 2018 from two dimensions of transformation efficiency (static) and productivity (dynamic) using the SBM-Undesirable model and the Malmquist–Luenberger productivity index. The results found that: First, there is still a potential for 21.7% improvement in the transformation efficiency of green innovation in pollution-intensive industries, the productivity is increasing and presents a dynamic evolution characteristic of "Λ" shape and industry heterogeneity exists in both the transformation efficiency and productivity. Second, if energy conservation and pollution emissions reduction are not considered, the transformation efficiency of green innovation will be underestimated by 6.3 percentage points and the productivity overestimated by 1.3 percentage points. Finally, pollution-intensive industries can improve green innovation performance from three paths: Unilateral, stepping and jumping. Based on the research conclusions, to better promote the green transformation of Chinese pollution-intensive industries, we recommend increased investment in scientific research to promote the application and promotion of green technologies; strengthen the level of supervision and management to flexibly make use of environmental regulations; and change the concept of policy implementation to explore the diversity and complementarity of green innovation policies.

**Keywords:**  pollution-intensive industry; green innovation; green innovation performance; SBM-undesirable model; Malmquist–Luenberger productivity index

## 1. Introduction

In the new era, green and innovation are the keys to driving industrial transformation and upgrading, and promoting economic quality and efficiency. At present, the imbalance between China's economic development, resources, and environment is becoming increasingly serious. The frequent occurrence of natural disasters such as air pollution, water pollution and soil salinization has not only brought serious economic losses, intensified social contradictions and harmed public health [1,2]. It also reveals the degree of deterioration of resources and environment caused by pollution from industry and the extreme vulnerability of ecosystems [3]. Severe problems of resources and environment have become the main bottleneck restricting the sustainable development of China's economy, which needs to be changed. The report of the 19th National Congress of the Communist Party of China points out that innovation is the primary driving force to lead development. Moreover, we should adhere to the harmonious coexistence of man and nature to promote green development. The "made in China 2025"

plan also emphasizes accelerating the green transformation and upgrading of manufacturing industry. Therefore, under the new era, industrialization innovation advocates promote the transformation and upgrading of China's industry and improvement of the quality and efficiency of the economy. It should be resource-saving, environment-friendly and sustainable green innovation. Green innovation will become an inevitable choice for sustainable development and competitive advantage of industries especially for pollution industries under the dual pressure of resources and environment.

Over a long time period, industry being the main body of China's national economy, an extensive economic growth model is not sustainable [4]. China's polluting industries are mainly distributed in the industrial sector, with high environmental sensitivity and great potential for green innovation. We consider it to be one of the principal causes of environmental pollution and ecological damage [5]. Green innovation is an effective means to break through the constraints of resources and environment and promote the sustainable development of industry. It has become the key to solving the problem of resources and environment and realizing the green development of polluting industries. So, how to accurately measure and improve the green innovation performance of polluting industries, and grasp the status quo of the green innovation performance of polluting industries has become a top priority for green transformation of polluting industries, and for sustainable development of the country or region.

Therefore, this study focuses on pollution-intensive industries. Based on the identification of pollution-intensive industries, it reasonably evaluates green innovation performance. Then it puts forward the path to improve green innovation performance. It aims to solve the following four problems: (1) How to identify which industries are pollution-intensive industries? (2) How do energy saving and emission reduction factors affect the innovation performance of pollution-intensive industries? (3) How to objectively and reasonably evaluate the performance of green innovation in pollution-intensive industries? (4) How to effectively improve the green innovation performance of pollution-intensive industries and promote the green transformation and sustainable development of China's pollution-intensive industries? It hopes to find a breakthrough and focus for China's industrial transformation and upgrading, and improvement of economic quality and efficiency. It is of great theoretical and practical significance for the Chinese government to plan reasonable green innovation policies and realize sustainable development.

## 2. Literature Review

### 2.1. Pollution-Intensive Industries

The research on pollution-intensive industry mainly focuses on connotation, industry scope, industrial transfer, innovation efficiency and capacity evaluation. The so-called pollution-intensive industry usually refers to the direct or indirect production of numerous pollutants in production activities. It will not only cause the deterioration of resources and environment but will also pose a threat to and impact on humans, animals and plants life and health [6]. Strictly, any industry produces more or less certain pollutants, while the pollution-intensive industries discharge more and more pollutants. They are the principal cause of resource and environmental pollution and ecosystem damage [5]. Effective identification of pollution-intensive industries can better guide the efficient investment of resource elements.

At present, there is no unified standard to define the scope of pollution-intensive industries. Academia mainly defines it through three methods: The first is to compare the cost of industry emission reductions and define the industries with a higher total cost as pollution-intensive industries [7,8], which is limited by data acquisition. The second is to compare the scale of industrial pollution discharge, and define the industries with higher proportion of total discharge of the sector as pollution-intensive industries [9,10]. Third, by comparing the intensity of industrial pollution discharge, it is determined that the industries with high level of pollutant discharge per unit economic output are pollution-intensive industries [11,12]. Methods two and three were adopted by most scholars because of easy access to data. Besides, in 2007, the State Council's "First National Pollution Source Census

Plan" clearly pointed out papermaking and paper products, agricultural and sideline food processing industries, and 11 other heavy pollution industries. The emission of main pollutants accounts for over 80% of the total emission of corresponding pollutants. In the "Second National Pollution Source Census Plan", in 2017, is listed all industrial activity units that produce "three wastes" as the object of inspection. Subsequently, in 2019, the "Second National Pollution Source Census Production and Emission Coefficient Manual (Trial Version)" was issued.

In addition, some scholars have also studied the transfer of pollution-intensive industries. They point out that the transfer direction of pollution-intensive industries tends to the areas with relatively loose environmental policies, emerging industrial markets and lower production costs [13,14]. So, the environment of such areas is more likely to be damaged [15]. Some scholars have also verified the phenomenon of pollutant transfer in the international industrial movement, which has a certain negative impact on the environment [16–18]. Research by Dam and Scholtens [19] shows that there is a tendency for multinational companies to move pollution-intensive industries to developing countries with weak environmental regulations. As for China, since the end of the 20th century, Shen et al. [20] and Hu et al. [21] have concluded that the eastern region and the Yangtze River Delta region have undertaken a lot of international industrial transfer. They have become the destinations of many pollution-intensive industries, such as the chemical industry, rubber, plastic, etc., with huge pollution emissions. At a later to the abovementioned studies, Dou et al. [22] found that pollution-intensive industries had shifted from the east to the central and western regions, and that pollutants such as wastewater and $SO_2$ had accompanied the move. In general, the scope of pollution-intensive industries varies in each country and region, and there is frequent industrial transfer accompanied by a large amount of pollutant emissions. It can be seen that a transformation and upgrading of pollution-intensive industries is imminent.

Regarding the evaluation of innovation efficiency or capacity of pollution-intensive industries, relevant researches show that the innovation efficiency or capacity of China's pollution-intensive industries is not optimistic, and there is still much room for improvement. Li et al. [23] measured the technological innovation capabilities of 11 heavily polluting industries from the perspective of input–output. The results show that, compared with developed countries, the technological innovation capabilities of various types of enterprises are not strong and a relative decline in technology has taken place, whether it is state-owned enterprises, private enterprises or foreign-funded enterprises. Based on the dual attributes of green innovation, Chao et al. [24] evaluated the green innovation efficiency of 16 heavy pollution industries, and judged that "innovation is not green" in China's heavy pollution industries. Lin et al. [25] evaluated the efficiency of green innovation in 28 manufacturing industries in China from 2006 to 2014. The study found that the overall efficiency of green innovation in manufacturing is relatively low, and the difference between industries is large, showing a wave-shaped dynamic trend. Fang et al. [26] constructed a DDF-DEA model to evaluate the green innovation efficiency of 17 heavily polluting industries. They believed that the green innovation efficiency of China's heavily polluting industries is generally low. Moreover, the entire industry is in the transitional phase of effective innovation rather than green innovation.

### 2.2. Green Innovation and Green Innovation Performance

The research on green innovation in foreign countries started earlier and began to prevail in domestic academic circles in 2005 [27]. Facing the increasing pressure of resources and environment, enterprises from all over the country are beginning to pay more attention to green innovation. The implementation of green innovation strategy is undoubtedly the only way to achieve sustainable development. The theory of Natural-Resource-Based View also proposed a model of sustainable economic and environmental development, emphasizing the importance of resources and the environment [28]. Moreover, green innovation is the effective integration of two seemingly contradictory paradigms: The mainstream social paradigm [29] and the new ecological paradigm [30]. At present, in general, research on green innovation is concentrated at four levels: National, regional, industry

and enterprise. Specifically, it is mainly carried out from the definition of the green innovation concept, research perspective, research content and main conclusions [31–33].

Generally, green innovation is also called ecological innovation, environmental innovation or sustainable innovation [34]. Different from traditional innovation, green innovation is an application activity to achieve economic growth, resource conservation and environmental protection with new ideas and technologies [27]. From the perspective of system theory, green innovation is the combination of industrial innovation system theory and green economy theory, involving green products and green processes [35]. It generally refers to introducing any new or significantly improved product, process, organizational change or marketing scheme to reduce the consumption of natural resources and the emission of harmful substances in the product life cycle [36]. Specific to the pollution industry, it means the optimal allocation of green innovation elements under the constraints of resources and environmental policies. Relevant research has also pointed out that the green innovation system is not only directly affected by resources and environmental policies. Besides, a policy will have an indirect impact on the system through external factors such as foreign direct investment and enterprise technology introduction. There are also large differences in the degree of impact between different companies [37]. The process of green innovation in polluting industries is a complex network system with multiple elements and paths interacting with each other [38]. Therefore, this study holds that green innovation of pollution industries is the sustainable innovation of balancing "economic quality and efficiency increase" and "reducing resource consumption and environmental pollution", based on the perspective of resources and the environment.

Concerning the research on green innovation performance, most scholars conduct research based on four perspectives, including environmental economics [39], innovation economics [40], strategic management [41] and industrial organization [42]. However, there is little literature to evaluate the performance of industrial green innovation from the industrial level. Tang et al. [43] investigated 496 industrial enterprises in China from 2002 to 2017. They found that mandatory and regulatory environmental regulations had a negative impact on the green innovation performance of small and medium-sized enterprises and enterprises in central and western regions. Zhang et al. [44] measured the efficiency of green innovation in Xi'an from 2003 to 2016 through the SBM-DDF model. It was found that, compared with mandatory and regulatory environmental regulations, market-based and voluntary regulations are more effective. Moreover, it had a non-linear inverted U-shaped relationship with green innovation efficiency. Li and Zeng [3] analyzed green innovation efficiency and influencing factors using the Super-SBM model based on the panel data of the pollution industry in China from 2011 to 2015. The results showed that there were obvious differences among 21 industries, among which manufacture and processing of ferrous metals were the weakest. Hu et al. [45] used panel data from 30 provinces in China from 2008 to 2017 to study the coordination effect of technology and institutional tools on green innovation performance. The study judged that the coordination of technical and institutional means has a significant impact on green innovation performance. Bi et al. [46] studied the impact of technology transfer of multinational companies on the performance of green innovation in China's manufacturing industry, taking 28 manufacturing industries in China from 2005 to 2011 as samples. It pointed out that the effect of the technology transfer of multinational companies on the performance of green innovation has a volatile development trend over time. Based on the perspective of technology transfer of multinational companies, Sui et al. [47] analyzed the influencing factors of the green innovation performance of the manufacturing industry. In contrast, the influence of technology spillovers of multinational companies is the highest, followed by the absorption capacity of the green innovation system, and then the social capital of the green innovation system.

The review of relevant literature shows that the previous literature was based on different research perspectives and methods to evaluate the performance of green innovation. It has important reference value for understanding and evaluating green innovation performance. However, there are also the following shortcomings: First, there is little literature that evaluates the performance of green innovation at the industrial level; second, most studies evaluate green innovation performance from

a single perspective of green innovation efficiency or capability; third, in the evaluation of green innovation performance, the relevant research lacks the consideration of the impact of energy saving and emission reduction factors on innovation performance; fourth, in the evaluation of the efficiency or capacity of green innovation in pollution-intensive industries, only the expected output in the development of pollution-intensive industries is considered, while the unintended output in the development of innovation is ignored.

Therefore, this study selects pollution-intensive industries as the research object, and defines the green innovation performance of pollution-intensive industries as the integration of green innovation transformation efficiency and green innovation productivity. On the basis of identifying pollution-intensive industries, the green innovation performance of pollution-intensive industries (2014–2018 panel data) in China is evaluated from the two dimensions of static (green innovation conversion efficiency) and dynamic (green innovation productivity). In the evaluation process, both the expected output and the undesired output are considered. In addition, it also analyzes the effect of energy saving and emission reduction factors on the innovation performance of pollution-intensive industries. Finally, based on the evaluation results, paths for improving the green innovation performance of pollution-intensive industries is proposed.

## 3. Materials and Methods

### 3.1. Identification of Pollution-Intensive Industries

To identify which industries are pollution intensive, this study referred to the method commonly used in most of the literature, that is, by calculating the pollution emission intensity [11,12]. The specific methods are as follows: First, calculate the pollution discharge value of each industrial pollutant unit output value; second, set the value range from 0 to 1, and standardize the pollutant emission value of each industrial pollution unit output value linearly. Then, add the weight of the above pollution discharge scores and calculate the average score of wastewater, waste gas and solid waste; third, summarize the calculated average score of "three wastes" and get the total emission intensity coefficient of the industry $\gamma^i$.

According to the method of calculating pollution emission intensity and based on relevant data of the "China Statistical Yearbook" and "China Environmental Statistics Yearbook" over the years, compared with the standards of "National Economic Industry Classification (GB/T4754-2002)" and "National Economic Industry Classification (GB/T4754-2011)". the identification results of pollution emission intensity of various industry were calculated, as shown in Table 1.

Table 1 shows the results of the division of industrial pollution emission intensity. This study divided the pollution emission intensity into three levels according to the practice: Severe, moderate and clean. The critical value setting principle is roughly divided according to 30% of the industrial pollution emission intensity value. Specifically: $\gamma^i \geq 0.152$ belongs to severe pollution industry, $0.017 \leq \gamma^i \leq 0.091$ belongs to moderate pollution industry, and $\gamma^i \leq 0.016$ belongs to clean industry. Taking industrial enterprises above designated size as the statistical caliber, based on the consistency of industrial classification and data availability, a total of 25 industries with severe pollution and moderate pollution were identified as pollution-intensive industries and as research objects.

In addition, according to the statistics of relevant data, the relevant indicators of pollution-intensive industries are calculated as follows: From 2014 to 2018, the proportion of elements invested in pollution-intensive industries in the industrial field is: The full-time equivalent of R&D personnel accounts for 68.97%; the internal expenditure of R&D funds accounts for 73.36%; the total energy consumption accounts for 94.90%. These investments create 66.34% of the effective invention patents; 75.13% of the sales revenue of new products are created for the industrial field, and 89.75% of the"three wastes"emissions are also generated at the same time. Overall, during the inspection period, the innovation input–output level of pollution-intensive industries is high, but is accompanied by a large number of pollutant emissions. So, how about the green innovation performance of

pollution-intensive industries? What is the impact of energy conservation and emission reduction on innovation performance of pollution-intensive industries? How to effectively improve the green innovation performance of pollution-intensive industries?

**Table 1.** Division results of industrial pollution emission intensity.

| Intensity Coefficient of Pollution Emission | Classification | Industries |
|---|---|---|
| $\gamma^i \geq 0.152$ | Severe pollution | Mining and washing of coal, mining of ferrous metal ores, mining of non-ferrous metal ores, processing of food from agricultural products, manufacture of textile, manufacture of paper and paper products, processing of petroleum, coking and processing of nucleus fuel, manufacture of chemical raw material and chemical products, manufacture and processing of ferrous metals, manufacture and processing of non-ferrous metals, manufacture of non-metallic mineral products, production and supply of electric power and heat power. |
| $0.017 \leq \gamma^i \leq 0.091$ | moderate pollution | Extraction of petroleum and natural gas, mining and processing of nonmetal ores, manufacture of foods, manufacture of liquor, beverages and refined tea, manufacture of textile, apparel and accessories, manufacture of leather, fur, feather and related products and shoes, manufacture of medicines, manufacture of chemical fiber, manufacture of rubber and plastic, manufacture of metal products, manufacture of motor vehicles, manufacture of railway equipment, ships, aerospace equipment and other transport equipment, manufacture of computer, communication and other electronic equipment. |
| $\gamma^i \leq 0.016$ | Clean | Manufacture of tobacco, processing of timbers and manufacture of wood, bamboo, rattan, palm and straw, manufacture of furniture, printing, reproduction of recording media, manufacture of artworks, and articles for culture, education, sports and recreation, manufacture of general-purpose machinery, manufacture of special purpose machinery, manufacture of electrical machinery and equipment, manufacture of measuring instrument and meter, Repairs service of metal products, machinery and equipment, production and distribution of gas, production and distribution of water. |

*3.2. Methods and Variables*

3.2.1. SBM-Undesirable Model

This study measured the transformation efficiency of green innovation of pollution-intensive industries using the SBM-Undesirable model. DEA is a relatively effective evaluation method based on input–output data. It has an absolute advantage in evaluating the relative efficiency of the same decision-making units. The traditional DEA models (such as CCR [48], BCC [49]) are based on the linear piecewise and radial theory to measure efficiency. They do not consider the relaxation of input and output, and ignore the existence of unexpected output. To make up for these defects, Tone [50] proposed the SBM-Undesirable model, put the relaxation variables directly into the aim function, and considered the invalid rate situation from the two perspectives of input and output. Its advantage lies in considering the input–output slack variable and the unexpected output. However, there are also shortcomings, which cannot solve the problem of comparing multiple effective decision-making units. Considering the importance of the pollution-intensive industry in the development of national economy, in this study, we evaluate the efficiency of green innovation transformation in pollution-intensive industries. We need to pay more attention to the expected output while considering the unexpected

output. Therefore, based on the SBM-Undesirable model proposed by Tone, this study constructs an SBM-Undesirable model with variable returns to scale from the perspective of output, as follows:

Each industry is regarded as a decision-making unit to construct the best production possibility boundary of each period. Suppose there are $n$ decision units; each decision unit has an input vector, which is recorded as $x \in R^m$, and two output vectors; the expected output vector is recorded as $y^g \in R^{s_1}$, the unexpected output vector is recorded as $y^b \in R^{s_2}$. Then the matrix can be defined as follows: $X = [x_1, \cdots, x_n] \in R^{m \times n}$, $Y^g = [y_1^g, \cdots, y_n^g] \in R^{s_1 \times n}$, $Y^b = [y_1^b, \cdots, y_n^b] \in R^{s_2 \times n}$, where, $X > 0$, $Y^g > 0$, $Y^b > 0$. Then, the production possibility set ($p$) is defined as $P = \{(x, y^g, y^b) | x \geq X\lambda, y^g \leq Y^g\lambda, y^b \geq Y^b\lambda, \sum \lambda = 1\}$. Then the SBM-Undesirable model can be expressed as:

$$
\min \rho = \frac{1}{1 + \frac{1}{s_1 + s_2}}(\sum_{r=1}^{s_1} \frac{s_r^g}{y_r^g} + \sum_{r=1}^{s_2} \frac{s_r^b}{y_r^b}), \text{ s.t.}
\begin{cases}
y_r^g - Y^g\lambda + s^g = 0 \\
y_r^b - Y^b\lambda - s^b = 0 \\
s^g \geq 0, s^b \geq 0, \sum \lambda = 1
\end{cases}
\tag{1}
$$

In Formula (1), $s, r, s_1, s_2$ are output slack variables, number of outputs, number of expected outputs and number of unexpected outputs, respectively; $\sum \lambda = 1$ means variable returns to scale. Objective function $\rho \in [0, 1]$, if and only if $\rho = 1$, the slack of each input of the decision unit is 0, which has complete efficiency.

In addition, this study measured the traditional innovation transformation efficiency of pollution-intensive industries using the SBM model from the perspective of output (remove $y^b > 0$ part), that is, it does not consider undesired output.

### 3.2.2. Malmquist–Luenberger Productivity Index

Based on the Malmquist–Luenberger productivity index, this study calculated the green innovation productivity of pollution-intensive industries. In 1992, Färe et al. [51] were first calculated using the Malmquist index. At the same time, the Malmquist index was decomposed into technical efficiency change and production technology change. The Malmquist index is based on the output distance function, which is used to evaluate and analyze the relationship among efficiency change, technological progress and technological efficiency. However, there is no workable solution in the VRS model. In 1997, Chung et al. [52] introduced the directional distance function to replace the output distance function, and proposed the Malmquist–Luenberger index to increase the expected output and reduce the unexpected output. However, there are some limitations in this method, and there is no workable solution when calculating the productivity change in the cross period. The specific steps of the Malmquist–Luenberger productivity index are as follows: First, construct the production possibility boundary of an economy by DEA method; second, calculate the distance between each decision unit and the production possibility boundary in the economy by using the directional distance function; third, calculate the Malmquist–Luenberger productivity index in this period based on the directional distance function of two periods.

First, consider the production possibility boundary of unexpected output. Set the unexpected output as the *X*-axis and the expected output as the *Y*-axis, as shown in Figure 1. Suppose there are three decision variables C, D and E and the input, expected output and unexpected output of the i decision unit are expressed by $x^i$, $y^i$ and $b^i$ respectively. The set of production possibilities can be expressed as: $P = \{(x^i, y^i, -b^i) : x \text{ product}(y^i, b^i)\}$. Among them, the expected output of the decision unit E is the largest, and a line passing through the point E parallel to the *X*-axis intersects the *Y*-axis at the point B. If the undesired output is not considered (i.e., the undesired output is in the "strong disposal" state), then the decision unit will not be constrained by the undesired output and will not produce the undesired output indefinitely. At this time, the most effective decision-making unit is E, the set of production possibilities is $\{0, y^E\}$, that is, the part between the linear BF and the *X*-axis in the first quadrant. At this time, the production possibility boundary is the maximum output $y^E$.

If the unexpected output is considered (i.e., the unexpected output is in the "weak disposal" state), then the expected output of decision units C and D is lower than that of E, because decision units C and D need to increase input to deal with this part of unexpected output. It can be seen that if the expected and undesired outputs are considered comprehensively, the productivity of decision units C and D may not necessarily be lower than E. According to the assumptions of monotonicity, convexity and comprehensive consideration of expected and undesired output, the set of production possibilities is the part between the envelope OCDEF and the *X*-axis. OCDEF is the boundary of production possibility at this time.

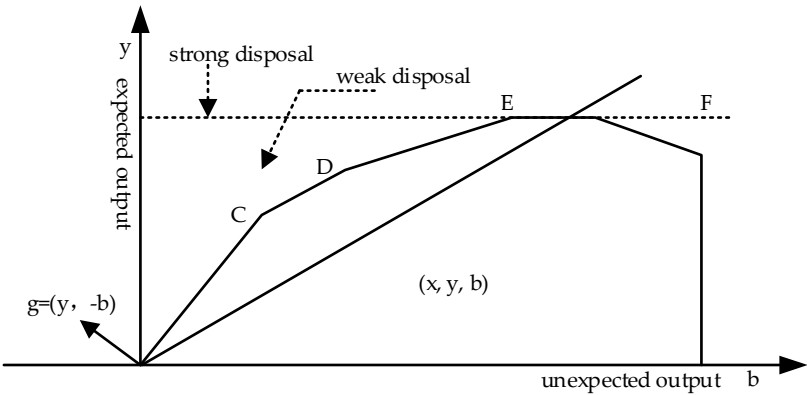

**Figure 1.** Schematic diagram of directional distance function (including unexpected output).

Second, environmental technology. In view of the objectivity of unexpected output, the production possibility set of environmental technology, as a technical structure that can comprehensively reflect expected and unexpected output [53], can be expressed as follows:

$$P(x) = \{(y, b) : x \, product(y, b)\}, x \in R^m, \tag{2}$$

In Formula (2), *p*(x) represents the set of possibilities of input x, expected output y and unexpected output b, among them, $x \in R^m$, $y \in R^{s_1}$, $x \in R^{s_2}$.

Third, the directional distance function. Based on the production possibility boundary, the distance between each decision unit and the production possibility boundary (i.e., relative efficiency) is calculated. The specific form of the directional distance function is:

$$\overrightarrow{D_0}(x, y, b; g) = \sup\{\beta : (y, b) + \beta g \in p(x)\}, \tag{3}$$

In Formula (3), $g = (g_y, -g_b)$ represents the direction vector of output expansion; suppose $g = (y, -b)$, that is, on the basis of the original stock, expected and unexpected output increase or decrease in proportion. In this case, the directional distance function represents that under the given input vector x, along the direction vector g, the maximum expansion multiple of the output vector $(y, -b)$ is β. The smaller the value of the directional distance function, the closer the production of the decision unit is to the boundary of production possibility, that is, the higher the productivity. If the value of the directional distance function is 0, then the decision unit is just on the boundary of the production possibility, and the production is invalid.

Finally, Malmquist–Luenberger productivity index. Referring to the research of Chung et al. [52], based on the directional distance function, the output-based Malmquist–Luenberger productivity index between the t and t + 1 period is defined as follows:

$$ML_t^{t+1} = \left\{ \frac{[1 + \vec{D}_0^t(x^t, y^t, b^t : g^t)]}{[1 + \vec{D}_0^t(x^{t+1}, y^{t+1}, b^{t+1} : g^{t+1})]} \times \frac{[1 + \vec{D}_0^{t+1}(x^t, y^t, b^t : g^t)]}{[1 + \vec{D}_0^{t+1}(x^{t+1}, y^{t+1}, b^{t+1} : g^{t+1})]} \right\}^{1/2}, \tag{4}$$

Then the Malmquist–Luenberger productivity index is divided into technical efficiency index (EFFCH) and technical progress index (TECH):

$$ML = EFFCH \times TECH, \tag{5}$$

$$EFFCH_t^{t+1} = \left\{ \frac{[1 + \vec{D}_0^t(x^t, y^t, b^t : g^t)]}{[1 + \vec{D}_0^t(x^{t+1}, y^{t+1}, b^{t+1} : g^{t+1})]} \right\}, \tag{6}$$

$$TECH_t^{t+1} = \left\{ \frac{[1 + \vec{D}_0^t(x^t, y^t, b^t : g^t)]}{[1 + \vec{D}_0^t(x^t, y^t, b^t : g^t)]} \times \frac{[1 + \vec{D}_0^{t+1}(x^{t+1}, y^{t+1}, b^{t+1} : g^{t+1})]}{[1 + \vec{D}_0^t(x^{t+1}, y^{t+1}, b^{t+1} : g^{t+1})]} \right\}^{1/2}, \tag{7}$$

In addition, this study measured the traditional innovation productivity of pollution-intensive industries using the Malmquist productivity index based on the output distance function, and the divided Malmquist productivity index into the technical efficiency index (EC) and technical progress index (TC).

### 3.2.3. Variable Description

After identifying pollution-intensive industries, this study evaluates the green innovation performance of pollution-intensive industries in China from 2014 to 2018. The following input–output variables are involved: The input variables include the full-time equivalent of R&D personnel, the internal expenditure of R&D funds and the total energy consumption from three aspects of human, capital and energy. The output variables mainly include three variables: The number of effective invention patents, the sales revenue of new products and the amount of industrial wastewater, waste gas and solid waste emissions. The selection of variables is based on the principles of scientificity, comprehensiveness and data availability. The specific meanings are as follows: (1) The full-time equivalent of R&D personnel and internal expenditure of R&D funds directly reflect the strength of human and capital investment in industrial R&D innovation; (2) the total energy consumption refers to the energy consumption capacity in the process of industrial green innovation activities; (3) the number of effective invention patents reflects the potential market revenue and economic value realization of industrial green innovation (the economic benefit of patent creation is directly related to the maintenance time of patent right. The longer the time is, the higher the market value will be. Therefore, this study chooses "the number of effective invention patents" to measure the potential market revenue and economic value realization of green innovation, and tries to make the research results close to the reality); (4) the sales revenue of new products is a supplementary indicator to measure the economic benefits of industrial green innovation and reflects the market acceptance ability of industrial innovation results; (5) the emission of industrial waste water, waste gas and solid waste, which can measure the environmental benefits brought by industrial green innovation. Among them, the internal expenditure of R&D funds adopts the perpetual inventory method [54] to estimate the capital stock and the constant price treatment in 2014. The sales revenue of new products shall be treated with constant price in 2014. The total energy consumption data comes from the "China Statistical Yearbook". The industrial "three wastes" data comes from the "China Environmental Statistics Yearbook". The rest of the data comes from the "China Science and Technology Statistics Yearbook" and statistical bulletins.

## 4. Empirical Analysis

### 4.1. Evaluation of Green Innovation Transformation Efficiency in Pollution-Intensive Industries

Based on the SBM-Undesirable model, this section analyzed the green innovation performance of pollution-intensive industries from the static dimension (transformation efficiency). The transformation efficiency values of green innovation in pollution-intensive industries in 2014–2018 are presented in Table 2.

**Table 2.** The transformation efficiency values of green innovation in pollution-intensive industries.

| Industry Classification | 2014 | 2015 | 2016 | 2017 | 2018 | Mean |
|---|---|---|---|---|---|---|
| Mining and washing of coal | 0.188 | 0.195 | 0.202 | 0.228 | 0.259 | 0.215 |
| Mining of ferrous metal ores | 1.000 | 1.000 | 1.000 | 1.000 | 1.000 | 1.000 |
| Mining of non-ferrous metal ores | 1.000 | 1.000 | 1.000 | 1.000 | 0.409 | 0.882 |
| Manufacture of paper and paper products | 0.484 | 0.483 | 0.598 | 0.605 | 0.578 | 0.550 |
| Processing of food from agricultural products | 0.545 | 0.527 | 0.474 | 0.456 | 0.512 | 0.503 |
| Manufacture of liquor, beverages and refined tea | 0.458 | 0.489 | 0.473 | 0.454 | 0.517 | 0.478 |
| Manufacture of foods | 0.594 | 0.576 | 0.552 | 0.565 | 0.572 | 0.572 |
| Manufacture of motor vehicles | 1.000 | 1.000 | 1.000 | 1.000 | 1.000 | 1.000 |
| Manufacture of railway equipment, ships, aerospace equipment and other transport equipment | 1.000 | 1.000 | 1.000 | 1.000 | 1.000 | 1.000 |
| Manufacture of computer, communication and other electronic equipment | 1.000 | 1.000 | 1.000 | 1.000 | 1.000 | 1.000 |
| Manufacture of textile | 1.000 | 1.000 | 0.724 | 0.582 | 0.486 | 0.758 |
| Processing of petroleum, coking and processing of nucleus fuel | 1.000 | 1.000 | 1.000 | 1.000 | 1.000 | 1.000 |
| Manufacture of non-metallic mineral products | 0.540 | 0.513 | 0.522 | 0.532 | 0.563 | 0.534 |
| Manufacture of chemical raw material and chemical products | 0.607 | 0.594 | 0.597 | 0.606 | 0.682 | 0.617 |
| Manufacture of chemical fiber | 1.000 | 1.000 | 1.000 | 0.681 | 1.000 | 0.936 |
| Manufacture and processing of ferrous metals | 0.475 | 0.505 | 0.519 | 0.619 | 1.000 | 0.624 |
| Manufacture and processing of non-ferrous metals | 1.000 | 0.917 | 0.821 | 0.838 | 1.000 | 0.915 |
| Manufacture of medicines | 0.634 | 0.627 | 0.621 | 0.601 | 0.678 | 0.632 |
| Production and supply of electric power and heat power | 1.000 | 1.000 | 1.000 | 1.000 | 1.000 | 1.000 |
| Extraction of petroleum and natural gas | 1.000 | 1.000 | 1.000 | 1.000 | 1.000 | 1.000 |
| Mining and processing of nonmetal ores | 1.000 | 1.000 | 1.000 | 1.000 | 1.000 | 1.000 |
| Manufacture of textile, apparel and accessories | 1.000 | 1.000 | 1.000 | 1.000 | 1.000 | 1.000 |
| Manufacture of leather, fur, feather and related products and shoes | 1.000 | 1.000 | 1.000 | 1.000 | 1.000 | 1.000 |
| Manufacture of rubber and plastic | 0.671 | 0.645 | 0.709 | 0.742 | 0.850 | 0.723 |
| Manufacture of metal products | 0.627 | 0.617 | 0.620 | 0.651 | 0.729 | 0.649 |
| Pollution-intensive industry | 0.793 | 0.788 | 0.777 | 0.766 | 0.793 | 0.783 |

In general, the transformation efficiency of green innovation in pollution-intensive industries is high, but there is still potential for improvement. Between 2014 and 2018, the average efficiency of green innovation transformation of pollution-intensive industries was 0.783, less than 1. This indicates that, under the input of existing factor resources (such as R&D personnel, funds, technology),

the efficiency of green innovation transformation of pollution-intensive industries in China still has 21.7% improvement potential.

In terms of different industries, the heterogeneity of pollution-intensive industries' green innovation transformation efficiency industries is obvious. Between 2014 and 2018, the average green innovation conversion efficiency of ten industries was 1. Including mining of ferrous metal ores, manufacture of motor vehicles, manufacture of railway equipment, ships, aerospace equipment and other transport equipment, manufacture of computers, communication and other electronic equipment, processing of petroleum, coking and processing of nucleus fuel, production and supply of electric power and heat power, extraction of petroleum and natural gas, mining and processing of nonmetal ores, manufacture of textile, apparel and accessories, and manufacture of leather, fur, feather and related products and shoes. This shows that the boundaries of green innovation production possibilities in these ten industries have reached the technological frontier. At this time, to continue to improve the transformation efficiency of green innovation, we must rely on strengthening management and innovative technologies to move the technological frontier out. The mean value of green innovation transformation efficiency of the remaining 15 industries during the inspection period is less than 1. It also shows that under the existing input of factor resources, the green innovation transformation efficiency of such pollution-intensive industries still has the potential to improve. Among them, the manufacture of chemical fiber is 0.936 (the potential for improvement is 6.4%), manufacture and processing of non-ferrous metals is 0.915 (8.5%), mining of non-ferrous metal ores is 0.882 (11.8%), manufacture of textile is 0.758 (24.2%), manufacture of rubber and plastic is 0.723 (27.7%), manufacture of metal products is 0.649 (35.1%), manufacture of medicines is 0.632 (36.8%), manufacture and processing of ferrous metals is 0.624 (37.6%), manufacture of chemical raw material and chemical products is 0.617 (38.2%), manufacture of foods is 0.572 (42.8%), manufacture of paper and paper products is 0.550 (45%), manufacture of non-metallic mineral products is 0.534 (46.6%), processing of food from agricultural products is 0.503 (49.7%), manufacture of liquor, beverages and refined tea is 0.478 (52.2%) and mining and washing of coal is 0.215 (78.5%). In a word, whether this kind of industry, or pollution-intensive industry as a whole. At this stage, we should stimulate the potential of green innovation to the greatest extent, and improve the transformation efficiency of green innovation through a series of means such as organizational structure optimization, management level improvement, and technological innovation promotion. Only by pushing the boundary of the best production possibility of green innovation towards the technological frontier can we continuously release huge innovation productivity.

In addition, it should be emphasized that in 15 industries where the average value of green innovation transformation efficiency is less than 1, the average value of green innovation transformation efficiency of 12 industries is lower than that of pollution-intensive industries (0.783). Among them, the green innovation transformation efficiency of the four industries, manufacture of paper and paper products, manufacture of non-metallic mineral products, processing of food from agricultural products, manufacture of liquor, beverages and refined tea, are all close to 50%, and the mining and washing of coal is as high as 78.5%. Therefore, we should pay more attention to the improvement of green innovation transformation efficiency in these five industries in the future.

In conclusion, during 2014–2018, the transformation efficiency of green innovation in pollution-intensive industries was generally high, with an average value of 0.783, and there was still potential for improvement. The average efficiency decreased continuously in the first four years to increase in 2018. The industry heterogeneity of green innovation transformation efficiency of pollution-intensive industry is obvious. Compared with similar studies in the past, Li and Zeng [3] calculated the green innovation efficiency of pollution-intensive industries based on the panel data of 2011–2015. The average efficiency shows the characteristics of dynamic fluctuation in the first four years to increase in 2015. There is a significant gap in the efficiency of green innovation among industries. The similarity of the research is that there are obvious differences among the industries of green innovation transformation efficiency. The difference is that the dynamic characteristics of

efficiency mean are not similar in two "five years". The major reason is the difference of unexpected output variables. Compared with "solid waste", this study considers that the results of selecting "three wastes" are closer to the reality.

*4.2. Evaluation of Green Innovation Productivity in Pollution-Intensive Industries*

This section analyzes the green innovation performance of pollution-intensive industries from the dynamic dimension based on the Malmquist–Luenberger productivity index. The values and decomposition of green innovation productivity in pollution-intensive industries from 2014 to 2018 are presented in Tables 3 and 4.

**Table 3.** The values and decomposition of green innovation productivity in pollution-intensive industries (by year).

| Years | Malmquist–Luenberger Productivity Index | | | Malmquist Productivity Index | | |
|---|---|---|---|---|---|---|
| | Technical Efficiency | Technical Progress | Green TFP | Technical Efficiency | Technical Progress | Traditional TFP |
| 2014~2015 | 0.999 | 1.051 | 1.050 | 1.013 | 1.029 | 1.042 |
| 2015~2016 | 0.986 | 1.113 | 1.096 | 0.969 | 1.153 | 1.117 |
| 2016~2017 | 0.997 | 1.037 | 1.034 | 1.004 | 1.072 | 1.077 |
| 2017~2018 | 1.014 | 0.984 | 1.008 | 1.083 | 0.931 | 1.008 |
| Mean | 0.999 | 1.046 | 1.047 | 1.016 | 1.043 | 1.060 |

**Table 4.** The values and decomposition of green innovation productivity in pollution-intensive industries (by industry).

| Industry Classification | Malmquist–Luenberger Productivity Index | | | Malmquist Productivity Index | | |
|---|---|---|---|---|---|---|
| | Technical Efficiency | Technical Progress | Green TFP | Technical Efficiency | Technical Progress | Traditional TFP |
| Mining and washing of coal | 1.001 | 0.999 | 1.000 | 1.095 | 0.970 | 1.062 |
| Mining of ferrous metal ores | 1.000 | 1.110 | 1.110 | 0.874 | 1.148 | 1.004 |
| Mining of non-ferrous metal ores | 0.884 | 1.132 | 1.011 | 0.935 | 1.064 | 0.995 |
| Manufacture of paper and paper products | 1.046 | 1.047 | 1.093 | 1.052 | 1.033 | 1.087 |
| Processing of food from agricultural products | 0.982 | 1.071 | 1.041 | 0.980 | 1.062 | 1.042 |
| Manufacture of liquor, beverages and refined tea | 1.035 | 1.044 | 1.075 | 1.047 | 1.029 | 1.077 |
| Manufacture of foods | 0.990 | 1.061 | 1.049 | 0.993 | 1.077 | 1.069 |
| Manufacture of motor vehicles | 1.000 | 1.026 | 1.026 | 1.000 | 1.008 | 1.008 |
| Manufacture of railway equipment, ships, aerospace equipment and other transport equipment | 1.000 | 1.016 | 1.016 | 0.997 | 1.036 | 1.032 |
| Manufacture of computer, communication and other electronic equipment | 1.000 | 1.108 | 1.108 | 1.000 | 1.084 | 1.084 |
| Manufacture of textile | 0.922 | 0.965 | 0.895 | 0.917 | 0.964 | 0.884 |
| Processing of petroleum, coking and processing of nucleus fuel | 1.000 | 1.040 | 1.040 | 1.000 | 1.042 | 1.042 |
| Manufacture of non-metallic mineral products | 1.031 | 1.006 | 1.037 | 1.030 | 1.019 | 1.049 |

**Table 4.** *Cont.*

| Industry Classification | Malmquist–Luenberger Productivity Index | | | Malmquist Productivity Index | | |
|---|---|---|---|---|---|---|
| | Technical Efficiency | Technical Progress | Green TFP | Technical Efficiency | Technical Progress | Traditional TFP |
| Manufacture of chemical raw material and chemical products | 1.027 | 1.071 | 1.092 | 1.035 | 1.041 | 1.078 |
| Manufacture of chemical fiber | 1.001 | 1.039 | 1.039 | 1.010 | 1.017 | 1.027 |
| Manufacture and processing of ferrous metals | 1.072 | 1.077 | 1.148 | 1.087 | 1.033 | 1.123 |
| Manufacture and processing of non-ferrous metals | 1.001 | 1.064 | 1.060 | 1.004 | 1.058 | 1.062 |
| Manufacture of medicines | 0.993 | 1.068 | 1.059 | 0.988 | 1.091 | 1.077 |
| Production and supply of electric power and heat power | 1.000 | 1.264 | 1.264 | 1.000 | 1.270 | 1.270 |
| Extraction of petroleum and natural gas | 1.000 | 1.052 | 1.052 | 1.250 | 1.120 | 1.400 |
| Mining and processing of nonmetal ores | 1.000 | 0.854 | 0.854 | 1.110 | 1.000 | 1.111 |
| Manufacture of textile, apparel and accessories | 1.000 | 1.026 | 1.026 | 0.974 | 0.968 | 0.943 |
| Manufacture of leather, fur, feather and related products and shoes | 1.000 | 0.996 | 0.996 | 0.990 | 0.970 | 0.961 |
| Manufacture of rubber and plastic | 1.017 | 1.023 | 1.039 | 1.054 | 1.003 | 1.057 |
| Manufacture of metal products | 1.049 | 1.001 | 1.048 | 1.044 | 1.016 | 1.061 |
| Pollution-intensive industry | 0.999 | 1.046 | 1.047 | 1.016 | 1.043 | 1.060 |

Overall, the green innovation productivity of pollution-intensive industries is increasing, and technological progress is the main driving mechanism. Between 2014 and 2018, the green innovation productivity of pollution-intensive industries was 1.047, with an average annual growth rate of 4.7%. Decomposing green innovation productivity can find that, during this period, the average annual growth rate of technological progress was 4.6%, which is the main driving mechanism for green innovation productivity, while the average annual growth rate of technological efficiency was −1%. This confirms the conclusion of the research on the transformation efficiency of green innovation of pollution-intensive industries, that is, pollution-intensive industries still have some potential to improve under the existing input of factor resources. We should improve the technical efficiency and promote the positive growth of technical efficiency by means of strengthening management and technological innovation. To realize the green innovation productivity growth of pollution-intensive industries from the single wheel drive mechanism of technological progress to the two-wheel drive mechanism of technological progress and technological efficiency improvement.

In terms of years, the green innovation productivity of pollution-intensive industries has a dynamic evolution characteristic of "Λ" shape. The growth rate of green innovation productivity of pollution-intensive industries increased from 5% in 2014–2015 to 9.6% in 2015–2016, then decreased to 3.4% in 2016–2017 and 0.8% in 2017–2018. By decomposing the green innovation productivity, it can be found that the growth rate of green innovation productivity decreased after the turning point in 2015–2016. The principal reason for this is that the compensation degree of technological progress to the negative growth of technology efficiency in 2015–2016 and 2016–2017 has decreased. The average annual growth rate of technological progress in 2015–2016 was 11.3%, and the average annual growth rate of technical efficiency was −1.4%, compared with 3.7% and −0.3% in 2016–2017. While the annual average growth rate of technological efficiency is 1.4% in 2017–2018, the negative growth rate of technological progress is −1.6%. Obviously, the improvement of technological efficiency cannot make up for the negative growth rate of technological progress. As a result, the growth rate of

green innovation productivity of pollution-intensive industries declined after the inflection point in 2015–2016. In addition, from 2014 to 2018, the technological progress of pollution-intensive industries is in a "Λ" shape, consistent with the dynamic evolution characteristics of green TFP, further illustrating that technological progress is the main mechanism driving the dynamic change of green innovation productivity of pollution-intensive industries.

In terms of industries, 21 pollution-intensive industries have the same growth and driving mechanism of green innovation productivity. Among them, green innovation productivity of 19 industries has achieved positive growth and technological progress is the primary driving mechanism. From 2014 to 2018, the green innovation productivity of the production and supply of electric power and heat power was the highest, with an average annual growth rate of 26.4%, the average annual growth rate of technological progress was 26.4%, and the contribution rate of technological progress reached 100%. The green innovation productivity of manufacture and processing of ferrous metals was second, with an average annual growth rate of 14.8%, an average annual growth rate of 7.7% and a contribution rate of 52%. The third was the manufacture of computer, communication and other electronic equipment, and the average annual growth rate of green innovation productivity was 10.8%, the average annual growth rate of technological progress was 10.8%, and the contribution rate of technological progress reached 100%.

In addition, it needs to be emphasized that, in the research on the transformation efficiency of green innovation of pollution-intensive industries, five industries need to be focused on. Including manufacture of paper and paper products, manufacture of non-metallic mineral products, processing of food from agricultural products, manufacture of liquor, beverages and refined tea, and mining and washing of coal. However, in the research of green innovation productivity, these five industries are not inferior. From 2014 to 2018, the green innovation productivity of the five industries achieved positive growth. Moreover, from the perspective of driving mechanism, the three industries have achieved double wheel drive of technical efficiency and technological progress. Including manufacture of paper and paper products, manufacture of non-metallic mineral products, manufacture of liquor, beverages and refined tea. Among them, the average annual growth rate of green innovation productivity in the manufacture of paper and paper products is 9.3%, the average annual growth rates of technical efficiency and technological progress are 4.6% and 4.7%, and the contribution rates are 49.5% and 50.5%, respectively. The annual growth rate of green innovation productivity of manufacture of non-metallic mineral products is 3.7%, the annual growth rate of technological efficiency and technological progress is 3.1% and 0.6%, respectively, and the contribution rate is 83.8% and 16.2%, respectively. The corresponding growth rates and contributions of the manufacture of liquor, beverages and refined tea were 7.9%, 3.5% and 4.4%, and 44.3% and 55.7%.

In conclusion, during 2014–2018, the overall green innovation productivity of pollution-intensive industries grew, and technological progress is the main driving mechanism. The green innovation productivity of pollution-intensive industries shows a dynamic evolution characteristic of "Λ". Compared with previous studies, Li et al. [55] found that the green innovation productivity of China's industrial sector increased slowly at an average annual rate of about 6% during 2001–2011, especially in emission-intensive industries. The majority of this growth comes from technological progress (57%). We can find that the average green innovation productivity of pollution-intensive industries has always maintained a growth trend. Technological progress is still the primary driving mechanism.

### 4.3. The Impact of Energy Conservation and Emission Reduction on Innovation Performance of Pollution-Intensive Industries

Based on the SBM model and the Malmquist productivity index, this section measured the innovation performance of pollution-intensive industries without considering the unexpected output from two dimensions of transformation efficiency (static) and productivity (dynamic) to investigate the impact of energy conservation and emission reduction on the innovation performance of pollution-intensive industries (Tables 3–5).

**Table 5.** The value of traditional innovation transformation efficiency of pollution-intensive industries.

| Industry Classification | 2014 | 2015 | 2016 | 2017 | 2018 | Mean |
|---|---|---|---|---|---|---|
| Mining and washing of coal | 0.138 | 0.146 | 0.152 | 0.173 | 0.196 | 0.161 |
| Mining of ferrous metal ores | 1.000 | 1.000 | 1.000 | 1.000 | 1.000 | 1.000 |
| Mining of non-ferrous metal ores | 1.000 | 1.000 | 1.000 | 1.000 | 0.365 | 0.873 |
| Manufacture of paper and paper products | 0.442 | 0.465 | 0.632 | 0.594 | 0.566 | 0.540 |
| Processing of food from agricultural products | 0.524 | 0.525 | 0.443 | 0.411 | 0.481 | 0.477 |
| Manufacture of liquor, beverages and refined tea | 0.415 | 0.464 | 0.432 | 0.406 | 0.491 | 0.441 |
| Manufacture of foods | 0.587 | 0.577 | 0.531 | 0.530 | 0.549 | 0.555 |
| Manufacture of motor vehicles | 1.000 | 1.000 | 1.000 | 1.000 | 1.000 | 1.000 |
| Manufacture of railway equipment, ships, aerospace equipment and other transport equipment | 1.000 | 1.000 | 1.000 | 0.831 | 0.835 | 0.933 |
| Manufacture of computer, communication and other electronic equipment | 1.000 | 1.000 | 1.000 | 1.000 | 1.000 | 1.000 |
| Manufacture of textile | 1.000 | 1.000 | 0.900 | 0.573 | 0.449 | 0.784 |
| Processing of petroleum, coking and processing of nucleus fuel | 1.000 | 1.000 | 1.000 | 1.000 | 1.000 | 1.000 |
| Manufacture of non-metallic mineral products | 0.537 | 0.501 | 0.512 | 0.520 | 0.563 | 0.526 |
| Manufacture of chemical raw material and chemical products | 0.645 | 0.625 | 0.627 | 0.638 | 0.760 | 0.659 |
| Manufacture of chemical fiber | 1.000 | 1.000 | 1.000 | 0.591 | 1.000 | 0.918 |
| Manufacture and processing of ferrous metals | 0.439 | 0.491 | 0.493 | 0.514 | 1.000 | 0.587 |
| Manufacture and processing of non-ferrous metals | 0.952 | 0.733 | 0.689 | 0.748 | 1.000 | 0.825 |
| Manufacture of medicines | 0.635 | 0.617 | 0.599 | 0.559 | 0.665 | 0.615 |
| Production and supply of electric power and heat power | 1.000 | 1.000 | 1.000 | 1.000 | 1.000 | 1.000 |
| Extraction of petroleum and natural gas | 0.023 | 0.086 | 0.128 | 0.154 | 0.142 | 0.107 |
| Mining and processing of nonmetal ores | 1.000 | 1.000 | 1.000 | 1.000 | 1.000 | 1.000 |
| Manufacture of textile, apparel and accessories | 1.000 | 1.000 | 0.746 | 0.463 | 0.627 | 0.767 |
| Manufacture of leather, fur, feather and related products and shoes | 1.000 | 1.000 | 1.000 | 1.000 | 1.000 | 1.000 |
| Manufacture of rubber and plastic | 0.519 | 0.509 | 0.566 | 0.596 | 0.725 | 0.583 |
| Manufacture of metal products | 0.605 | 0.593 | 0.613 | 0.628 | 0.744 | 0.636 |
| Pollution-intensive industry | 0.739 | 0.733 | 0.722 | 0.677 | 0.726 | 0.720 |

In terms of transformation efficiency, the transformation efficiency of green innovation in pollution-intensive industries is significantly higher than that of traditional innovation (Tables 2 and 5). Between 2014 and 2018, the average value of traditional innovation transformation efficiency of pollution-intensive industries (without considering energy conservation and emission reduction factors) was 0.720, which is about 6.3 percentage points lower than the average value of green innovation transformation efficiency (considering energy conservation and emission reduction factors) of 0.783. It can be seen that if energy conservation and emission reduction factors are not considered, the innovation transformation efficiency of pollution-intensive industries will be underestimated. Some studies also point out that environmental regulation policy has a significant negative impact on the innovation efficiency of China's heavily polluting industries [26]. However, in terms of different industries, there is industry heterogeneity within pollution-intensive industries. During the inspection period, the average value of green innovation transformation efficiency (0.758, 0.617) of

the textile industry, the chemical raw material industry and the chemical product manufacturing industry is lower than that of traditional innovation transformation efficiency (0.784, 0.659). What are the reasons? Through further combing the data, we found that these two industries have more energy consumption and pollutant emissions in pollution-intensive industries. From 2014 to 2018, the total energy consumption of pollution-intensive industries (25 industries) was 13,921,462,100 tons of standard coal, and the total emission of "three wastes" was 132,471,415,000 tons. Among them, the total energy consumption of manufacture of textile, manufacture of chemical raw material and chemical products is 2,802,585,600 tons of standard coal, and the total emission of "three wastes" is 24,666.08 million tons. The total energy consumption and the total emission of "three wastes" account for 20% and 19% of the total pollution-intensive industry, respectively. Therefore, for the high energy consumption and pollution industries such as manufacture of textile, manufacture of chemical raw material and chemical products, we should focus on energy conservation and emission reduction. Mostly from the aspects of management level improvement and clean technology innovation, etc., it is important to promote the transformation efficiency of green innovation to make it higher than the transformation efficiency of traditional innovation.

In terms of productivity, the productivity of green innovation in pollution-intensive industries is significantly lower than that of traditional innovation (Tables 3 and 4). Between 2014 and 2018, the annual growth rate of traditional innovation productivity of pollution-intensive industries (without considering energy conservation and emission reduction factors) was 6%, which was about 1.3 percentage points higher than the annual growth rate of green innovation productivity (considering energy conservation and emission reduction factors) of 4.7%. In this way, if we do not consider the factors of energy conservation and emission reduction, we will overestimate the innovation productivity of pollution-intensive industries. Similar to related studies, Li and Lin [56] found that, if the impact of carbon emissions is ignored, the innovation productivity of China's manufacturing industry will be overestimated. However, in terms of industries, there is industry heterogeneity in pollution-intensive industries. During the inspection period, the green innovation productivity of 14 pollution-intensive industries, accounting for more than half of the total number of pollution-intensive industries (25 industries), was lower than that of traditional industries. Further combing the data, it can be concluded that the total energy consumption and "three wastes" emissions of 14 industries (7,093,314,000 tons of standard coal and 6,502,353,000 tons), respectively, account for 51% and 49% of the corresponding total amount of pollution-intensive industries. It can be seen that energy conservation and emission reduction play an important role in the development of pollution-intensive industries.

In conclusion, to evaluate the innovation performance of pollution-intensive industries scientifically, we must consider the factors of energy conservation and emission reduction in industrial development. In this way, we can use objective and effective research conclusions to guide the transformation and upgrading of pollution-intensive industries, healthy development and economic quality and efficiency improvement in China.

### 4.4. The Path to Improve Green Innovation Performance of Pollution-Intensive Industries

This study evaluated the green innovation performance of pollution-intensive industries from two dimensions of transformation efficiency (static) and productivity (dynamic). It is found that the transformation efficiency of green innovation in pollution-intensive industries is high, and there is still potential for improvement. At the same time, the green innovation productivity of pollution-intensive industries is increasing, and the driving mechanism is the single wheel drive of technological progress. In addition, the transformation efficiency and productivity of green innovation in pollution-intensive industries have industry heterogeneity. Therefore, the green innovation performance of pollution-intensive industries in China still has a certain potential to improve. So, which pollution-intensive industries should we focus on? How to effectively improve the green innovation performance of pollution-intensive industries?

In view of this, the transformation efficiency of green innovation in pollution-intensive industries (Table 2) is taken as the horizontal axis, and the green innovation productivity (Table 4) is taken as the vertical axis to construct the path map for improving the performance of green innovation in pollution-intensive industries (see Figure 2). At the same time, taking the mean value of green innovation transformation efficiency (0.783) and the mean value of green innovation productivity (1.047) as the boundary, it is defined as high if greater than the mean value, and low if less than the mean value. According to the high and low values of the two, it is divided into four combinations: A, B, C and D. That is, "high, low", "high, high", "low, high" and "low, low". Different combination trends in the future represent different promotion paths.

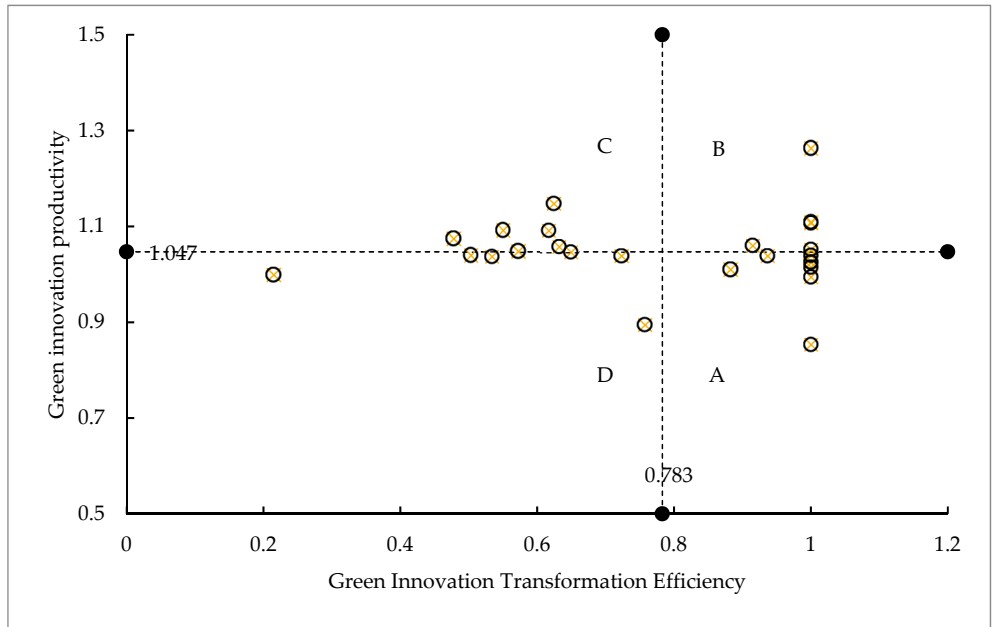

**Figure 2.** Paths for improving green innovation performance in pollution-intensive industries.

Among them, group A represents the state of "high green innovation conversion efficiency and low green innovation productivity". The group includes eight industries, including mining of non-ferrous metal ores, manufacture of motor vehicles, manufacture of railway equipment, ships, aerospace equipment and other transport equipment, processing of petroleum, coking and processing of nucleus fuel, manufacture of chemical fiber, mining and processing of nonmetal ores, manufacture of textile, apparel and accessories, manufacture of leather, fur, feather and related products and shoes. Group B is in a "high and high" state, which includes five industries, i.e., mining of ferrous metal ores, manufacture of computer, communication and other electronic equipment, manufacture and processing of non-ferrous metals, production and supply of electric power and heat power and extraction of petroleum and natural gas. Group C is in a "low and high" state, which includes seven industries: Manufacture of paper and paper products, manufacture of liquor, beverages and refined tea, manufacture of foods, manufacture of chemical raw material and chemical products, manufacture and processing of ferrous metals, manufacture of medicines and manufacture of metal products. Group D is in a "low, low" state, which includes five industries: mining and washing of coal, processing of food from agricultural products, manufacture of textile, manufacture of non-metallic mineral products, manufacture of rubber and plastic.

Among groups A, B, C and D, group B, which is in the position of "high green innovation transformation efficiency and high green innovation productivity", is obviously the optimal combination. It is the target orientation of improving the green innovation performance of pollution-intensive industries. Based on this, this study provides three paths to promote the green innovation performance of pollution-intensive industries to group B: First, the unilateral

promotion path of A→B or C→B. That is, in the development process of pollution-intensive industries, the industries with relatively low green innovation transformation efficiency or productivity are taken as a breakthrough point to strengthen management and improve performance. Second, the stepping promotion path of D→A→B or D→C→B. That is, it is mainly aimed at industries in the "low and low" state. At this time, priority should be given to the development of industries with comparative advantages, while making up for the industries with comparative disadvantages, so as to realize the transition D→A or D→C, and then through the unilateral promotion path to B, finally achieve the overall improvement of green innovation performance. Third, the jumping promotion path of D→B, that is, directly from the "low, low" state to the "high, high" state. Generally, such a way of promotion is difficult to achieve. However, under the guidance of relevant national policies, the strengthening of management within the industry and the continuous promotion of technological innovation, such jumping promotion path still has a certain possibility of practical operation.

## 5. Conclusions and Implications

### 5.1. Conclusions

Based on the identification of pollution-intensive industries in the industrial field, this study analyzed the innovative development characteristics of pollution-intensive industries, and then evaluated the green innovation performance of these industries from two dimensions of transformation efficiency (static) and productivity (dynamic). The study also analyzed the industry distribution, the dynamic evolution characteristics and the driving mechanism of green innovation performance of pollution-intensive industries in 2014–2018. In addition, based on the investigation of energy consumption and environmental pollution factors, this study analyzed the impact of energy conservation and emission reduction on the innovation performance of pollution-intensive industries. Finally, we proposed three ways to improve the green innovation performance of pollution-intensive industries. Through these analyses, the following conclusions are obtained:

1.  In the industrial field, 25 industries, such as mining and washing of coal and manufacture of metal products, are pollution-intensive industries. The innovation input–output level of pollution-intensive industries is high, but accompanied by a large number of pollutant emissions.
2.  The transformation efficiency of green innovation in pollution-intensive industries is relatively high, with 21.7% improvement potential, and the heterogeneity of the transformation efficiency of green innovation in these industries is obvious. In these industries, manufacture of paper and paper products, manufacture of non-metallic mineral products, processing of food from agricultural products, manufacture of liquor, beverages and refined tea, and mining and washing of coal need to be focused on.
3.  The green innovation productivity of pollution-intensive industries is increasing positively, showing a dynamic evolution characteristic of "Λ" and technological progress is the main driving mechanism.
4.  During the inspection period, without considering the factors of energy conservation and emission reduction, the transformation efficiency of innovation in pollution-intensive industries will be underestimated by 6.3 percentage points, and the productivity will be overestimated by 1.3 percentage points.
5.  Considering that the green innovation performance of pollution-intensive industries still has a certain potential to improve, we provide three improvement paths: Unilateral, stepping and jumping from the two dimensions of transformation efficiency and productivity.

### 5.2. Implications

In view of the above conclusions, the improvement of Chinese pollution-intensive industries' green innovation performance needs to start from the following aspects:

1.  Increase investment in scientific research and promote the application and promotion of green technology. Green technology is the core of promoting the green transformation of pollution-intensive industries. We should focus on providing research and development support for clean production, recycling and other energy-saving and emission reduction technologies. Actively guide R&D investment to incline to special green technologies for pollution-intensive industries. Realize the wide application and promotion of green technology in pollution-intensive industries, to speed up the green transformation of pollution-intensive industries.

2.  Strengthen the level of supervision and management, and flexibly use the means of environmental regulation. Environmental regulation is the key factor to promote the technological innovation of pollution-intensive industries. On the one hand, we should continue to strengthen the enforcement of environmental regulations and improve the management mechanisms of industry approval, pollution discharge, cleaner production audit and industry elimination. Promote the green innovation performance of pollution-intensive industries with mechanism binding force. On the other hand, we should distinguish the order type and incentive type environmental regulation tools, and combine them flexibly to stimulate the technological innovation of industrial pollution control and the innovation of the management system. To improve the green innovation performance of pollution-intensive industries, we should reduce the cost of pollution control, improve the energy utilization rate and reduce pollution emissions.

3.  Change the concept of policy implementation and explore diversified and complementary ways of green innovation policies. Green innovation policy is the target orientation of sustainable development of pollution-intensive industries. The effectiveness of relevant policies, such as environmental regulation, market cultivation and government support, varies with different industries, so targeted policy complementarity can really become an important means to force the green innovation of pollution-intensive industries. Generally, the improvement of green innovation performance of pollution-intensive industries depends on the combination of strict environmental regulation policies and strong government support policies, supplemented by excellent research investment and research personnel. The improvement of green innovation performance of small-scale pollution intensive enterprises is more inclined to strict environmental regulation policies and active market cultivation policies, supplemented by excellent R&D investment.

**Author Contributions:** Conceptualization, C.W. and J.L.; Methodology, C.W.; Software, C.W.; Data curation, C.W.; Formal analysis, C.W.; Investigation, C.W. and J.L.; Supervision, C.W. and J.L.; Resources, J.L.; Writing—Original draft preparation, C.W.; Writing—Review and editing, J.L. All authors have read and agreed to the published version of the manuscript.

**Funding:** This research was funded by Key Project of Philosophy and Social Science Research of the Ministry of EducatioN, grant number 15JZD021" and "Science and Technology Plan Project of Tianjin, grant number 18ZLZDZF00190".

**Acknowledgments:** The authors would like to acknowledge the professionals who collaborated during this study. I would also like to thank the editor and the anonymous referees at the *Sustainability* for insightful comments.

**Conflicts of Interest:** The authors declare no conflict of interest.

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
