# Peer review of "The Evaluation and Promotion Path of Green Innovation Performance in Chinese Pollution-Intensive Industry"

_sustainability, doi:10.3390/su12104198_

Round 1

Reviewer 1 Report

The article contains the evaluation of the green innovation performance in pollution intensive industries by using static and dynamic approach. Although the scope of the study concerns the case of China it can be interesting and useful for a wider range of readers.  

Strengths

The aim, scope of the study and applied methods are defined well. The literature review justifies the research topic. Methods used in the research are detailed described.

The obtained results were clearly presented and well interpreted. In my opinion they would be more valuable if they were compared to results from other surveys in different countries.

Conclusions and recommendations correspond to the conducted analysis.

Weaknesses

The article structure is not fully adapted to the editorial requirements. The part “Introduction” contains a mix of literature review and methods used in the study. It should be separated.

The title “Industry Identification, Model Construction and Variable Description” should be changed to “Materials and methods”. The reasonable division into two subsections should be maintained.

After ordering the structure, the article is recommended for publication.

Author Response

Responses to Reviewer 1 Comments

We are much grateful for your careful reading of our manuscript and your valuable comments and suggestions to help improve the paper. We have now carefully revised the paper in light of all the comments and suggestions. The following is a point-by-point response. Your original comments are shown in italics for easy reference. Please kindly note that our corresponding amendments in the revised paper have been highlighted in yellow for your easy reference.

Due to the adoption of revision mode to revise the manuscript, in order to avoid confusion of line numbers, the line numbers in the following responses are the final version, that is, the line numbers after the revision are accepted.

Point-Strengths 1:

The article contains the evaluation of the green innovation performance in pollution intensive industries by using static and dynamic approach. Although the scope of the study concerns the case of China it can be interesting and useful for a wider range of readers. 

Strengths

The aim, scope of the study and applied methods are defined well. The literature review justifies the research topic. Methods used in the research are detailed described.

The obtained results were clearly presented and well interpreted. In my opinion they would be more valuable if they were compared to results from other surveys in different countries.

Conclusions and recommendations correspond to the conducted analysis.

Response 1:

We thank you for the positive remarks of our work. Following the opinion, in future research, we will increase the value of research by focusing on “compared to results from other surveys in different countries”.

Point-Weaknesses 2:

The article structure is not fully adapted to the editorial requirements. The part “Introduction” contains a mix of literature review and methods used in the study. It should be separated.

The title “Industry Identification, Model Construction and Variable Description” should be changed to “Materials and methods”. The reasonable division into two subsections should be maintained.

After ordering the structure, the article is recommended for publication.

Response 2:

We thank you for these helpful comments.

First, according to your suggestions, we put the discussion on the concept of green innovation in the “Introduction” into the section of the literature review “2.2 Green Innovation and Green Innovation Performance”, and rearrange the contents of the section 2.2 (Please see lines 145-158). At the same time, the part of “selection of research objects and proposal of research problems” in the “Introduction” was rectified in terms of logic and language, making the research more rigorous and appropriate (Please see lines 51-73).Avoid that the part “Introduction” contains a mix of literature review and methods used in the study.

Second, according to your suggestion, we change the title “3. Industry identification, model construction, and variable description” to “3. Materials and Methods”. And we divided the section 3 into two parts, of which 3.1 is “Identification of Pollution-Intensive Industries”, 3.2 is “Methods and Variables”. (Please see line 208, line 244).

Reviewer 2 Report

  1. The title shows“The Evaluation and Promotion Path”; however, it is not clear about it in the content.
  2. The research gap is not clear.
  3. How did the author define “ Green Innovation and Green Innovation Performance?”
  4. There is no notation for the formulation. Therefore, it is not clearly to understand the input, output, and output slack variables.
  5. In table 2, what is the maximum value? Where did the author get the data? 
  6. The contribution is also not clearly to be justified.
  7. Based on the current content, it is not clear for the content.

Author Response

Responses to Reviewer 2 Comments

We are much grateful for your careful reading of our manuscript and your valuable comments and suggestions to help improve the paper. We have now carefully revised the paper in light of all the comments and suggestions. The following is a point-by-point response. Your original comments are shown in italics for easy reference. Please kindly note that our corresponding amendments in the revised paper have been highlighted in yellow for your easy reference.

Due to the adoption of revision mode to revise the manuscript, in order to avoid confusion of line numbers, the line numbers in the following responses are the final version, that is, the line numbers after the revision are accepted.

Point 1:

  1. The title shows “The Evaluation and Promotion Path”; however, it is not clear about it in the content.

Response 1:

We thank you for this helpful comment.

Sorry for the confusion in the article title and content. In the revised manuscript, the structure of the article has been adjusted according to the requirements of the journal. The evaluation and promotion path of green innovation performance shows in the section 4, which is divided into three parts and they are “4.1. Evaluation of Green Innovation Transformation Efficiency in Pollution-intensive Industries; 4.2. Evaluation of Green Innovation Productivity in Pollution-intensive Industries; 4.4. The Path to Improve Green Innovation Performance of Pollution-Intensive Industries”. (Please see line 361, line 424, line 559).

Point 2:

  1. The research gap is not clear.

Response 2:

We thank you for this helpful comment.

We are very sorry for the confusion caused by insufficient expression of the research gap. In order to solve this problem, in the literature review part of the article, we have revised some expressions and pointed out some gaps in the existing research. (please see lines 187-197). The purpose of this article is to fill up this research gap by analyzing the evaluation and promotion path. In addition, we have highlighted the innovation of this article. (please see lines 198-207).

The innovation of this article is:1) Based on the industrial level, we choose pollution-intensive industries.2) We define the concept of green innovation performance. We evaluate the performance of green innovation from the two dimensions of static (transformation efficiency) and dynamic (productivity). It provides a new research idea for evaluating the performance of green innovation.3) In the evaluation of green innovation performance, we consider the impact of energy saving and emission reduction factors on innovation performance.4)Fourth, based on the two dimensions of green innovation performance evaluation, the article puts forward innovative ways to improve the green innovation performance of pollution-intensive industries.

Point 3:

  1. How did the author define “Green Innovation and Green Innovation Performance?”

Response 3:

We thank you for this helpful comment.

Green Innovation:green innovation is the sustainable innovation of balancing “economic quality and efficiency increase” and “reducing resource consumption and environmental pollution”, based on the perspective of resources and the environment. (please see lines 158-161).

Green Innovation Performance: green innovation performance is the integration of green innovation transformation efficiency (static dimension) and green innovation productivity (dynamic dimension). (please see lines198-200).

Point 4:

  1. There is no notation for the formulation. Therefore, it is not clearly to understand the input, output, and output slack variables.

Response 4:

We thank you for this helpful comment.

First, Sorry for the formula problem. According to your suggestion, we infer your question refers to formula (1).

Second, In the revised version, we want to clarify that formula (1) is an SBM-Undesirable model with variable returns to scale from the perspective of output. It evolved from the non-radial and non-angular SBM-Undesirable model [50].

Third, we list the formula and explain the relevant variables:

The non-radial and non-angular SBM-Undesirable model:

See attached for formula

Among, x is the input vector, yg is the expected output vector, yb is the unexpected output vector;

s- is the input relaxation variable, s is the output relaxation variable, r is the number of outputs s1 is the number of expected outputs s2 is the number of unexpected outputs

On this basis, this study constructs an SBM-Undesirable model with variable returns to scale from the perspective of output is constructed, as follows:

See attached for formula

Among, yg is the expected output vector, yb is the unexpected output vector; s is the output relaxation variable, r is the number of outputs s1 is the number of expected outputs s2 is the number of unexpected outputs. (please see lines 262-272).

Reference:

  1. Tone, K. A slacks-based measure of super-efficiency in data envelopment analysis. Eur. J. Oper. Res. 2001, 143, 32–41. DOI: 10.1016/S0377-2217(01)00324-1

Point 5:

  1. In table 2, what is the maximum value? Where did the author get the data?

Response 5:

We thank you for this helpful comment.

First, we would like to clarify this question, from the perspective of different industries, the maximum value of the mean value of green innovation transformation efficiency of pollution-intensive industries is 1. In terms of years, the Average of green innovation transformation efficiency of pollution-intensive industries, maximum value is 0.793. (please see line 366, Table 2).

Point 6:

  1. The contribution is also not clearly to be justified.

Response 6:

We thank you for this helpful comment.

According to your suggestion, in the revised manuscript, we have highlighted the innovation of this article. Make the research more persuasive. (please see lines 198-207).

Point 7:

  1. Based on the current content, it is not clear for the content.

Response 7:

We thank you for this helpful comment.

Sorry for the confusion caused by the research content of this article. According to your suggestion, in the revised manuscript, we adjusted the title of some sections of the article, pointed out the research gap and the innovation of the article.

In addition, we also simplify the lengthy sentences in the full text, making the overall logical thinking of the article clearer and improving the quality of the article.

Please see the yellow part of the text for the specific modification.

Reviewer 3 Report

Dear Authors,

Thank you of the opportunity to review your manuscript entitled The Evaluation and Promotion Path of Green Innovation Performance in Chinese Pollution Intensive Industry.

The topic of the paper is extremely interesting. Some suggestions I placed below. I really like how you explained the reason why the research has been conducted. Overall the research problem is clearly explained, and the research questions placed at line 81 are very concrete.

The main suggestion is to work a little bit about discussion. I know that your approach is innovate; however, even some short comparisons with existing studies will highlight your impact.

Others (technical suggestions to the authors)

The line number of the comments refer to the line numbers added by the MDPI System.

Some sentences are too long sentences. Please see e.g. (much more you will find in the manuscript – you must go through it again and simplify them):

51. Different from the traditional innovation, green innovation is an application activity entrusted with new ideas and new technologies to achieve resource conservation and environmental protection expectations, to achieve coordinated development of economy, resources, and the environment, and to obtain corresponding economic benefits [4].

68. Considering the importance of efficient use of resources, energy, and environmental pollution prevention, the green innovation performance discussed in this study is a comprehensive performance that combines the transformation efficiency and productivity of green innovation.

87. In response to the above four …. (please start this sentence from the new line).

  1. Why ‘25 Chinese pollution-intensive industries’. Could you clarified?

97-103. This is not necessary – I recommend to delete it.  

  1. Please use mathematical signs: You have m*n and it should be m x n (please check the whole manuscript in that matter).

Well done!

Author Response

Responses to Reviewer 3 Comments

We are much grateful for your careful reading of our manuscript and your valuable comments and suggestions to help improve the paper. We have now carefully revised the paper in light of all the comments and suggestions. The following is a point-by-point response. Your original comments are shown in italics for easy reference. Please kindly note that our corresponding amendments in the revised paper have been highlighted in yellow for your easy reference.

Due to the adoption of revision mode to revise the manuscript, in order to avoid confusion of line numbers, the line numbers in the following responses are the final version, that is, the line numbers after the revision are accepted.

Point-Strengths 1:

The topic of the paper is extremely interesting. Some suggestions I placed below. I really like how you explained the reason why the research has been conducted. Overall, the research problem is clearly explained, and the research questions placed at line 81 are very concrete.

Response 1:

We thank you for the encouraging comments of our work and we are greatly appreciative.

Point -Weaknesses 2:

The main suggestion is to work a little bit about discussion. I know that your approach is innovate; however, even some short comparisons with existing studies will highlight your impact.

Response 2:

We thank you for these helpful comments.

According to your suggestion, in the revised manuscript, we added the comparison with the previous research in the discussion part of the article and listed the relevant literature as the support. (Please see lines 411-423, lines 492-500, lines 518-519, lines 544-546).

Reference:

  1. Li, D., & Zeng, T. Are China’s intensive pollution industries greening? An analysis based on green innovation efficiency. J. Clean Prod. 2020, 259, 120901. DOI: 10.1016/J.JCLEPRO.2020.120901
  2. Li, H., Zhang, J., Osei, E., & Yu, M. Sustainable development of China’s industrial economy: An empirical study of the period 2001–2011. Sustainability 2018, 10(3), 764. DOI: 10.3390/SU10030764
  3. Li, K., & Lin, B. Impact of energy conservation policies on the green productivity in China’s manufacturing sector: Evidence from a three-stage DEA model. Appl. Energy 2016, 168, 351–363. DOI: 10.1016/J.APENERGY.2016.01.104

Point 3:

Others (technical suggestions to the authors)

The line number of the comments refer to the line numbers added by the MDPI System.

Some sentences are too long sentences. Please see e.g. (much more you will find in the manuscript – you must go through it again and simplify them):

  1. Different from the traditional innovation, green innovation is an application activity entrusted with new ideas and new technologies to achieve resource conservation and environmental protection expectations, to achieve coordinated development of economy, resources, and the environment, and to obtain corresponding economic benefits [4].
  2. Considering the importance of efficient use of resources, energy, and environmental pollution prevention, the green innovation performance discussed in this study is a comprehensive performance that combines the transformation efficiency and productivity of green innovation.

Response 3:

We thank truly the you for your detailed comments and helpful suggestions to help improve our paper.

First, according to your suggestion, we have revised sentences 51 and 68:

  1. Different from traditional innovation, green innovation is an application activity to achieve economic growth, resource conservation, and environmental protection with new ideas and technologies. (Please see lines 145-147).
  2. Sorry for choice delete this sentence. We need to explain it to you here, because we have perfected the “Introduction” logically and linguistically again, and finally deleted sentence 68.

Second, according to your suggestion, we reread the original manuscript and simplify the sentences in the article. Please kindly note that our corresponding amendments in the revised paper have been highlighted in yellow for your easy reference.

Point 4:

  1. In response to the above four …. (please start this sentence from the new line).

Response 4:

We thank you for this helpful comment.

Here need to clarify, because we have made a changed the “introduction” section to improve the article quality. In the revised manuscript, we changed the sentence to “…And then put forward the path to improve its green innovation performance. It aims to solve the following four problems: (1) How to identify pollution-intensive industries? …”. And the paragraph appears as a separate paragraph. (Please see lines 62-66).

Point 5:

  1. Why ‘25 Chinese pollution-intensive industries’. Could you clarified?

Response 5:

We thank you for these helpful comments.

First, we are sorry for the confusion in the selection of subjects. We would like to clarify is that the basis for choosing 25 pollution-intensive industries is: first, calculate the carbon emission intensity coefficient of various industries of China industry [11][12]. Second, according to the Convention, we divide the pollution emission intensity into three levels: heavy, medium and clean. The principle of setting the critical value is roughly divided according to 30% of the proportion of industrial pollution emission intensity coefficient. Third, 25 industries which belong to heavy pollution and moderate pollution are regarded as pollution-intensive industries. (Please see lines 210-231).

Reference:

  1. Mani, M., & Wheeler, D. In Search of Pollution Havens? Dirty Industry in the World Economy, 1960 to 1995. J. Environ. Dev. 1998, 7, 215–247. DOI: 10.1177/107049659800700302
  2. Qiu, F. D., Jiang, T., Zhang, C. M., & Shan, Y. B. Spatial relocation and mechanism of pollution-intensive industries in Jiangsu Province. Sci. Geogr. Sin. 2013, 33, 789-796. DOI: 10.13249/j.cnki.sgs.2013.07.78907.x

Point 6:

97-103. This is not necessary – I recommend to delete it. 

Response 6:

We thank you for these helpful comments.

According to your suggestion, we have deleted this part in the manuscript revision.

Point 7:

  1. Please use mathematical signs: You have m*n and it should be m x n (please check the whole manuscript in that matter).

Response 7:

We thank you for these helpful comments.

According to your suggestion, we checked the whole manuscript and changed all “*” to the mathematical signs "×”. (Please see line 266).

Reviewer 4 Report

Please include the limitations of the chosen approach and method.

List all the things that the model does not address and encompass.

Please explain are there any uncertainties related to the data sources (how reliable are statistics?).

Implications: 1) why R&D is promoted?, 2) address law enforcement and 3) address the details of how policy implementation should be done.

Author Response

Responses to Reviewer 4 Comments

We are much grateful for your careful reading of our manuscript and your valuable comments and suggestions to help improve the paper. We have now carefully revised the paper in light of all the comments and suggestions. The following is a point-by-point response. Your original comments are shown in italics for easy reference. Please kindly note that our corresponding amendments in the revised paper have been highlighted in yellow for your easy reference.

Due to the adoption of revision mode to revise the manuscript, in order to avoid confusion of line numbers, the line numbers in the following responses are the final version, that is, the line numbers after the revision are accepted.

Point 1:

Please include the limitations of the chosen approach and method.

Response 1:

We thank you for this helpful comment.

First, we need to explain that the methods adopted in this article are SBM-Undesirable model and Malmquist-Luenberger productivity index. SBM-Undesirable model is used to measure the transformation efficiency of green innovation in pollution-intensive industries. Malmquist-Luenberger productivity index is used to measure the green innovation productivity of pollution-intensive industries.

Second, according to your suggestion, in the manuscript's revision, we added a detailed discussion of the two research methods, and discussed the limitations of the research methods. (please see lines247-261, lines278-286).

Reference:

  1. Tone, K. A slacks-based measure of super-efficiency in data envelopment analysis. Eur. J. Oper. Res. 2001, 143, 32–41. DOI: 10.1016/S0377-2217(01)00324-1
  2. Färe, R., Grosskopf, S., Lindgren, B., & Roos, P. Productivity Changes in Swedish Pharamacies 1980-1989: A Non-Parametric Malmquist Approach. J. Prod. Anal.1992, 3, 85–101. DOI: 10.1007/BF00158770
  3. Chung, Y. H., Färe, R., & Grosskopf, S. Productivity and Undesirable Outputs: A Directional Distance Function Approach. J. Environ. Manage. 1997, 51, 229–240.DOI:10.1006/JEMA.1997.0146

Point 2:

List all the things that the model does not address and encompass.

Response 2:

We thank you for this helpful comment.

According to your suggestion, we would like to explain, SBM-Undesirable Model cannot solve the problem of comparing multiple effective decision-making units. (please see lines254-256). The Malmquist-Luenberger productivity index has no workable solution when calculating the productivity change in the cross period. (please see lines285-286).

Point 3:

Please explain are there any uncertainties related to the data sources (how reliable are statistics?).

Response 3:

We thank you for this helpful comment.

Our original data comes from the authoritative database of China, namely “Chinese Statistical Yearbook”, “China Environmental Statistics Yearbook”, “China Science and Technology Statistics Yearbook” which is widely used in the research of many scholars and has high recognition and reliability.

Point 4:

Implications: 1) why R&D is promoted? 2) address law enforcement and 3) address the details of how policy implementation should be done.

Response 4:

We thank you for these helpful comments.

About this question, we would like to explain that the Implications part of the article is based on the research conclusion of the article. The results of the study show that during 2014-2018, technological progress is the primary driving mechanism for green innovation performance in pollution-intensive industries.

Therefore,

1) We want to clarify that the article refers to increasing investment in scientific research. In fact, it emphasizes that we should actively promote the research and development of special green technologies and actively guide the research and development resources to incline to green technologies. This is consistent with the conclusion that technological progress is the primary driving mechanism. (please see lines648-654).

2) According to your suggestion, in the manuscript’s revision, we have improved the content of the manuscript and proposed more specific measures and means to solve the law enforcement problem. (please see lines655-665).

3) For details of policy implementation, in the manuscript’s revision, we focused on the detailed discussion, improved this part of the content, is the article enlightenment part become more reference value and guiding significance. (please see lines666-677).

Round 2

Reviewer 2 Report

All the questions are already addressed.  I have no more problems.